# Efficient computation of adjoint sensitivities at steady-state in ODE models of biochemical reaction networks

**Polina Lakrisenko**[1,2], **Paul Stapor**[1,2], **Stephan Grein**[3], **Łukasz Paszkowski**[4], **Dilan Pathirana**[3], **Fabian Fröhlich**[5], **Glenn Terje Lines**[4], **Daniel Weindl**[1], **Jan Hasenauer**[1,3]*

**1** Computational Health Center, Helmholtz Zentrum München Deutsches Forschungszentrum für Gesundheit und Umwelt (GmbH), Neuherberg, Germany, **2** Center for Mathematics, Technische Universität München, Garching, Germany, **3** University of Bonn, Life and Medical Sciences Institute, Bonn, Germany, **4** Simula Research Laboratory, Oslo, Norway, **5** Department of Systems Biology, Harvard Medical School, Boston, Massachusetts, United States of America

* jan.hasenauer@uni-bonn.de

**Data Availability Statement:** All the presented approaches have been integrated into AMICI

## Abstract

Dynamical models in the form of systems of ordinary differential equations have become a standard tool in systems biology. Many parameters of such models are usually unknown and have to be inferred from experimental data. Gradient-based optimization has proven to be effective for parameter estimation. However, computing gradients becomes increasingly costly for larger models, which are required for capturing the complex interactions of multiple biochemical pathways. Adjoint sensitivity analysis has been pivotal for working with such large models, but methods tailored for steady-state data are currently not available. We propose a new adjoint method for computing gradients, which is applicable if the experimental data include steady-state measurements. The method is based on a reformulation of the backward integration problem to a system of linear algebraic equations. The evaluation of the proposed method using real-world problems shows a speedup of total simulation time by a factor of up to 4.4. Our results demonstrate that the proposed approach can achieve a substantial improvement in computation time, in particular for large-scale models, where computational efficiency is critical.

## Author summary

Large-scale dynamical models are nowadays widely used for the analysis of complex processes and the integration of large-scale data sets. However, computational cost is often a bottleneck. Here, we propose a new gradient computation method that facilitates the parameterization of large-scale models based on steady-state measurements. The method can be combined with existing gradient computation methods for time-course measurements. Accordingly, it is an essential contribution to the environment of computationally efficient approaches for the study of large-scale screening and omics data, but not tailored

version 0.11.32, an open-source tool for efficient simulation and sensitivity analysis (https://github.com/AMICI-dev/AMICI). All code and data to reproduce the analysis has been deposited at https://doi.org/10.5281/zenodo.6963596.

**Funding:** This work was supported by the European Union's Horizon 2020 research and innovation program (CanPathPro, https://cordis.europa.eu/project/id/686282; Grant No. 686282; P. S., G.T.L., L.P.), the German Federal Ministry of Education and Research (BMBF, https://www.bmbf.de) within the e:Med funding scheme (junior research alliance PeriNAA, grant no. 01ZX1916A; D.W., P.L.), the Deutsche Forschungsgemeinschaft (DFG, German Research Foundation, https://www.dfg.de/, Project ID 432325352, 443187771; S.G, D. P; EXC 2047 390873048, EXC 2151 390685813; J. H.), the Human Frontier Science Program (https://www.hfsp.org/, Grant no. LT000259/2019-L1; F. F.), and the National Cancer Institute (https://www.cancer.gov, Grant no. U54-CA225088; F.F.). The funders had no role in study design, data collection and analysis, decision to publish, or preparation of the manuscript.

**Competing interests:** The authors have declared that no competing interests exist.

to biological applications, and, therefore, also useful beyond the field of computational biology.

This is a *PLOS Computational Biology* Methods paper.

## Introduction

Ordinary differential equation (ODE) models are widely used to describe the dynamics of biochemical processes such as signalling [1–3], metabolism [4, 5] or gene regulation [6, 7]. These models can capture the mechanistic details of interactions between biochemical species, aggregate current knowledge and integrate heterogeneous data types. However, such models often possess a large number of parameters. While some parameter values can possibly be extracted from databases such as BRENDA [8] or SABIO-RK [9], others usually need to be inferred from problem-specific experimental data. The complexity of this inference problem depends on the model size, the number of unknown parameters and data availability.

Data availability and, in general, the structure and information content of data sets differs substantially between projects. On the experimental side, one often distinguishes between time-course and dose-response data. On the computational side, different experimental setups correspond to different types of simulation setups (Fig 1a): (i) the initial conditions of the simulation can either be specified explicitly as numerical values or parameters, or implicitly with a steady-state constraint; and (ii) either a dynamic phase has to be simulated or only a steady-state has to be determined. The experimental setup and the corresponding formulation of the simulation can be encoded using formats, such as the Simulation Experiment Description Markup Language (SED-ML) [10] or the Parameter Estimation tables (PEtab) [11].

In many studies, there is some data available regarding the system's steady-state [12–15]. There are two distinct cases: (1) the system is assumed to start in a steady state, then it is perturbed and enters a dynamic state; or (2) the system is assumed to start in a dynamic state and after some time it reaches a steady state (Fig 1a). These two cases will be referred to as pre- and post-equilibration, respectively. The system can start in a steady state and reach a steady state again after a perturbation, therefore, the two cases are not mutually exclusive.

Pre- and post-equilibration as part of the simulation complicate parameter estimation and other subsequent tasks as the numerical computation of steady states of nonlinear systems is challenging. Therefore, a variety of computational methods have been proposed to derive analytical expressions for steady states. These methods rely, for example, on graph theory [16, 17], py-substitution [18] and some can even take positivity constraints into account [19]. Yet, for a large number of application problems these analytical approaches are not applicable and one needs to resort to numerical approaches. Fortunately, steady states can be computed without time-course simulations [20, 21]. Nowadays, variants of Newton's method are widely used for equilibration. Tailored implementations of Newton's method achieve a speedup of up to 100-fold compared to steady-state calculation via numerical simulation [21].

Efficient numerical simulation and steady-state calculation are cornerstones for parameter estimation [22]. However, several studies found that the availability of the objective function gradient is also highly beneficial for the parameterization for dynamical modelling (see, e.g. [23, 24]). Indeed, gradient-based multi-start local and global optimization methods seem to often outperform gradient-free optimization methods [23].

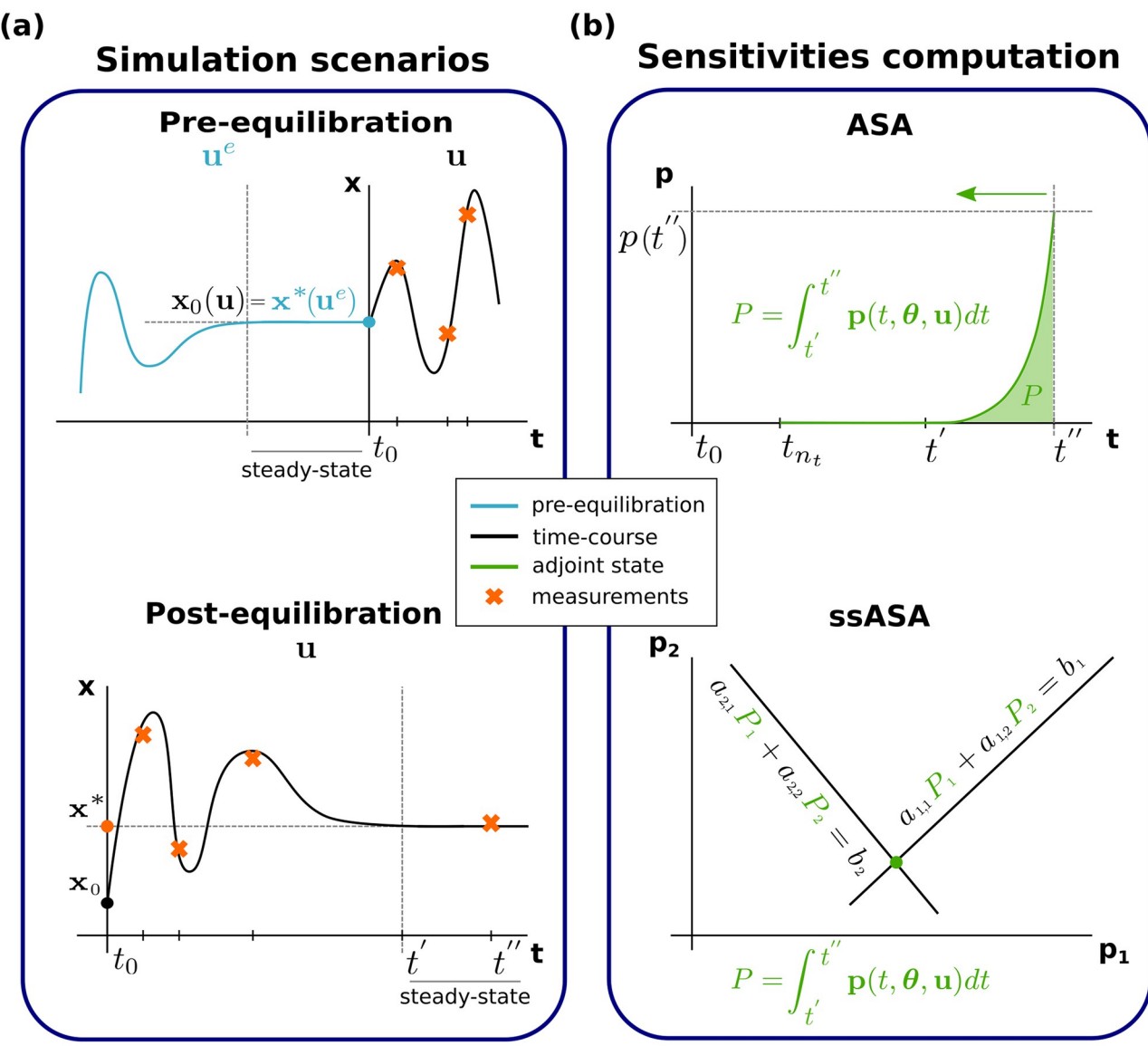

**Fig 1. Steady-state simulation scenarios, and computation methods for ASA.** (a) Different scenarios requiring computation of the model steady state. Top: The pre-equilibration case, where the system is at steady state at the beginning of the experiment. This means that by $t = t_0$ the system has reached its steady state ($x^*$) under some condition ($u^e$) (blue line). At $t = t_0$ the system was perturbed and measurements were taken at time points $t_j > t_0$ (orange crosses). The pre-equilibration steady state ($x^*$) is the initial state under the experimental conditions ($u$), i.e. $x_0(u) = x^*$. Bottom: The post-equilibration case, where at some point after the beginning of the experiment, the system reaches its steady state ($x^*$) and measurements for this time point ($t = t''$) are available (orange cross at $t = t''$). Measurements of the transient state may also be available (orange crosses in $t < t''$). (b) Alternative approaches for computing sensitivities. Top: The standard ASA approach that requires numerical integration until convergence to the steady state and subsequent backward integration of the adjoint state ($p$) ODE on the same time interval. Bottom: The proposed adjoint method that circumvents backward numerical integration and requires solving a system of linear algebraic equations instead.

Objective function gradients can be computed using automatic differentiation (see, e.g., [25]), finite differences, and forward or adjoint sensitivity analysis [26]. While approximations obtained using finite differences are often numerically unreliable [24], forward and adjoint methods allow for an accurate assessment [26]. All sensitivity analysis methods are, in principle, available for pre- and post-equilibration. With forward sensitivity analysis, one can simulate state sensitivity ODEs along with the dynamical system until approximate convergence to

a steady state. This comes at a high computational cost and scales poorly with the number of states and parameters, as one has to simulate a system of ODEs with the dimension $(n_x + 1)n_\theta$. The existing adjoint method for pre- and post-equilibration simulates the model until approximate convergence to a steady state, and then solves the corresponding backward equation for the same time interval [12, 27]. This is computationally costly as well but possesses, in contrast to the forward method, improved scalability; here, one has to sequentially simulate two systems of the size $n_x$. However, the forward method can be easily adapted to exploit steady-state constraints. One can find state sensitivities at steady state without numerical integration, by solving a system of linear equations per parameter (see, e.g., [20]). The downside is that this approach is not applicable for systems with singular Jacobians. A corresponding approach for adjoint methods is currently missing.

In this paper, we introduce a novel adjoint method for the computation of objective function gradients for problems with steady-state constraints. The proposed method exploits the steady-state constraint to circumvent numerical backward integration for pre- and post-equilibration. Instead, a system of equations is solved to compute parts of the objective function gradient. We provide an implementation of the method in the open-source toolbox AMICI and investigate its accuracy and efficiency on three real-world problems. Our study demonstrates that the new method achieves a substantial speedup for gradient computation, which is particularly valuable for large models.

## Methods

In this section, we introduce the considered classes of mathematical models and parameter estimation problems. Subsequently, we describe the established method for gradient calculation via adjoint sensitivity analysis (ASA) and propose a new formulation for the steady-state case. We discuss how the method can be applied to models of biochemical reaction networks. The method, however, is not restricted to the field of computational biology. It can be used for any model of the form described in this section with pre- or post-equilibration, or both.

### Mathematical modeling and simulation

We consider ODE models of the form

$$\dot{\mathbf{x}} = \mathbf{f}(\mathbf{x}(t), \boldsymbol{\theta}, \mathbf{u}), \ \mathbf{x}(t_0) = \mathbf{x}_0(\boldsymbol{\theta}, \mathbf{u}), \tag{1}$$

in which $\mathbf{x}(t, \boldsymbol{\theta}, \mathbf{u}) \in \mathbb{R}_+^{n_x}$ is the vector of state variables, $\boldsymbol{\theta} \in \mathbb{R}_+^{n_\theta}$ is the parameter vector, $\mathbf{u} \in \mathbb{R}_+^{n_u}$ is the vector of constant inputs and $t \in \mathbb{R}$ is the time. The vector field of the ODE model is $\mathbf{f} : \mathbb{R}_+^{n_x} \times \mathbb{R}_+^{n_\theta} \times \mathbb{R}_+^{n_u} \to \mathbb{R}_+^{n_x}$, which is Lipschitz-continuous with respect to x, and the initial condition at $t_0$ is $\mathbf{x}_0 : \mathbb{R}_+^{n_\theta} \times \mathbb{R}_+^{n_u} \to \mathbb{R}_+^{n_x}$. The ODE model determines the time evolution of the state variables, given the parameters. In case of biochemical reaction networks, this can be e.g., concentrations of biochemical species and reaction rate constants, respectively. The inputs ($\mathbf{u}$) describe experimental conditions, e.g., regime of administration of a drug or composition of a cell culture medium. In many models, the state variables approach a steady-state $\mathbf{x}^*(\boldsymbol{\theta}, \mathbf{u})$ for large values of $t$:

$$\mathbf{x}^*(\boldsymbol{\theta}, \mathbf{u}) = \lim_{t \to \infty} \mathbf{x}(t, \mathbf{x}_0(\boldsymbol{\theta}, \mathbf{u}), \boldsymbol{\theta}, \mathbf{u}). \tag{2}$$

In this paper we assume that for each initial condition ($\mathbf{x}_0(\boldsymbol{\theta}, \mathbf{u})$) there exists an exponentially stable steady state. Both pre- and post-equilibration cases assume existence of a steady state that the dynamical system can reach fast enough to appropriately describe the modeled

biochemical process. This assumption is often implicitly made when working with steady-state data.

Additionally, we denote by $\mathbf{u}^e$ the inputs corresponding to a pre-equilibration condition, if pre-equilibration is required. In this case, the initial condition is specified as a steady state corresponding to pre-equilibration condition $(\mathbf{x}(t_0) = \mathbf{x}^*(\boldsymbol{\theta}, \mathbf{u}^e))$, i.e. as a steady state of

$$\dot{\mathbf{x}} = \mathbf{f}(\mathbf{x}(t, \boldsymbol{\theta}, \mathbf{u}^e), \boldsymbol{\theta}, \mathbf{u}^e), \ \ \mathbf{x}(t_0, \boldsymbol{\theta}) = \mathbf{x}_0(\boldsymbol{\theta}, \mathbf{u}^e).$$

The solution of the ODE system (1) is usually not available in closed form. Therefore, numerical simulation algorithms are widely used to determine the time-course of state variables. Here, we employ an algorithm based on the backward-differentiation formula (BDF) to study the—often stiff—ODE models of biochemical processes. For the computation of steady states of ODE models (2) analytical methods, numerical simulation, and numerical equation solvers (e.g. Newton's method) can be employed [16–19, 21]. Here, we employ numerical integration until time derivatives $\dot{\mathbf{x}}$ become sufficiently small. For example, one can run numerical integration until the condition

$$\sqrt{\frac{1}{n_x}\sum_{i=1}^{n_x}(\dot{x}_i w_i)^2} < 1, \quad \text{where } w_i = \frac{1}{\text{rtol} * x_i + \text{atol}} \tag{3}$$

is fulfilled, where "rtol" and "atol" denote relative and absolute tolerances, respectively.

## Experimental data

Experiments provide information about observable components of the biochemical processes, which are in general subsets or functions of the state variables ($\mathbf{x}$). The dependence of the observables $\mathbf{y}(t, \boldsymbol{\theta}, \mathbf{u}) \in \mathbb{R}^{n_y}$ on the state variables and parameters is modelled as

$$\mathbf{y}(t, \boldsymbol{\theta}, \mathbf{u}) = \mathbf{h}(\mathbf{x}(t, \boldsymbol{\theta}, \mathbf{u}), \boldsymbol{\theta}, \mathbf{u}), \tag{4}$$

with output map $\mathbf{h} : \mathbb{R}^{n_x}_+ \times \mathbb{R}^{n_\theta}_+ \times \mathbb{R}^{n_u}_+ \to \mathbb{R}^{n_y}$. As measurements are corrupted by noise, we model them as noise-corrupted realizations of the observables. Here, we consider two types of measurements:

1. Time-course measurements with independent, normally-distributed noise,

$$\bar{y}_{ij} = y_i(t_j, \boldsymbol{\theta}, \mathbf{u}) + \varepsilon_{ij} \ \ \text{with } \varepsilon_{ij} \sim \mathcal{N}(0, \sigma_{ij}^2), \tag{5}$$

in which $i = 1, \ldots, n_y$ and $j = 1, \ldots, n_t$ index observables and time points, and $\sigma_{ij}^2 \in \mathbb{R}_+$ denotes the noise variance. We assume without loss of generality that $t_j < t_{j+1}$ and denote the collection of these time-course measurements by $\mathcal{D}_t$.

2. Independent, normally distributed measurements of the steady state,

$$\bar{y}_{i*} = y_{i*}(\boldsymbol{\theta}, \mathbf{u}) + \varepsilon_{i*} \quad \text{with } \varepsilon_{i*} \sim \mathcal{N}(0, \sigma_{i*}^2), \tag{6}$$

in which $y_{i*}(\boldsymbol{\theta}) = \mathbf{h}(\mathbf{x}^*(\boldsymbol{\theta}, \mathbf{u}), \boldsymbol{\theta}, \mathbf{u})$ denotes the values of the observable $y_i$ at the steady state $\mathbf{x}^*(\boldsymbol{\theta}, \mathbf{u})$, and $\sigma_{i*}^2 \in \mathbb{R}_+$ denotes the noise variance. For the case of multi-stable systems, the state for the starting point of the post-equilibration needs to be considered. We denote the collection of these measurements in steady state by $\mathcal{D}_*$.

Measurements from multiple experiments may be available, which can be described by a set of inputs $\mathcal{U} = \{\mathbf{u}_1, \ldots, \mathbf{u}_n\}$. Further in this section, without loss of generality, we assume that measurements from one experiment, described by $\mathbf{u}$ are available.

**Remark**: For the subsequent sections we assume, without loss of generality, that the noise is normally distributed and the variance is known. The results also apply to unknown noise variances and other noise distributions, given that measurements are independent and the mean of the noise is equal to zero.

## Parameter estimation

As the parameters of biochemical processes are often difficult to determine experimentally, they are typically unknown. Hence, the values of the parameters ($\boldsymbol{\theta}$) have to be estimated from experimental data. This is usually achieved using frequentist or Bayesian inference.

Frequentist inference focuses on maximum likelihood estimates (MLEs) and the assessment of their uncertainties. MLEs maximize the likelihood function ($\mathcal{L}$), which is a measure for the distance between measurement and simulation, and this is equivalent to minimizing the negative log-likelihood function ($\mathcal{J}$). For independent, normally-distributed measurements, the negative log-likelihood function is given by

$$
\begin{aligned}
\mathcal{J}(\boldsymbol{\theta}) = -\log \mathcal{L}(\boldsymbol{\theta}) = \quad & \frac{1}{2} \sum_{i=1}^{n_y} \sum_{j=1}^{n_t} \left( \log\left(2\pi\sigma_{ij}^2\right) + \left( \frac{\bar{y}_{ij} - y_i(t_j, \boldsymbol{\theta}, \mathbf{u})}{\sigma_{ij}} \right)^2 \right) \\
& + \frac{1}{2} \sum_{i=1}^{n_y} \left( \log\left(2\pi\sigma_{i*}^2\right) + \left( \frac{\bar{y}_{i*} - y_{i*}(\boldsymbol{\theta}, \mathbf{u})}{\sigma_{i*}} \right)^2 \right),
\end{aligned}
\tag{7}
$$

in which the first part accounts for the time-course measurements ($\mathcal{D}_t$) and the second part for the steady state measurements ($\mathcal{D}_*$). The MLE is computed by solving the optimization problem

$$
\boldsymbol{\theta}^* = \arg \min_{\boldsymbol{\theta}} \ \mathcal{J}(\boldsymbol{\theta}).
$$

The parameters are often constrained, e.g. to be non-negative, and might be log-transforms of the original model parameters in order to improve convergence of the optimizer [24, 28].

Bayesian inference focuses on the assessment of the posterior distribution, which encodes the information provided by the data as well as prior knowledge. In practice the first step is usually to compute the maximum *a posteriori* estimator. This is achieved by minimizing the (often unnormalized) negative log-posterior, yielding an optimization problem similar to the one encountered in frequentist inference [29].

Parameter estimation for ODE models often involves objective functions (e.g. negative log-likelihood or negative log-posterior functions) that are non-convex and possess local minima. To solve these parameter estimation problems, global optimization methods are used. A globalization strategy that has proven to be efficient is multi-start local optimization with gradient-based local optimization methods [23, 24]. The computation time of these methods is usually dominated by the computation time required for the evaluation of the objective function gradient. For the objective function (7), the gradient is given by

$$
\left. \frac{\partial \mathcal{J}}{\partial \theta_k} \right|_{\boldsymbol{\theta}} = -\sum_{i=1}^{n_y} \sum_{j=1}^{n_t} \frac{(\bar{y}_{ij} - y_i(t_j, \boldsymbol{\theta}, \mathbf{u}))}{\sigma_{ij}^2} \left. \frac{\partial y_i}{\partial \theta_k} \right|_{t_j, \boldsymbol{\theta}} - \sum_{i=1}^{n_y} \frac{(\bar{y}_{i*} - y_{i*}(\boldsymbol{\theta}, \mathbf{u}))}{\sigma_{i*}^2} \left. \frac{\partial y_{i*}}{\partial \theta_k} \right|_{\boldsymbol{\theta}}, \tag{8}
$$

in which $\partial y_i / \partial \theta_k$ denotes the sensitivity of the observable $y_i$ with respect to the parameter $\theta_k$ at a specific time point $t_j$ or in steady state.

## Adjoint sensitivity analysis (ASA)

For large-scale ODE models, the objective function gradient for time-course data $\mathcal{D}_t$ is usually calculated using ASA. This approach has a long history and relates to Green's function method of sensitivity analysis (see e.g. [30] and references therein) and is used across scientific disciplines. [26] adopted it to parameter estimation problems in biological applications for the case of time-course measurements without pre- or post-equilibration. For a description of ASA, readers are referred to [26]. This approach circumvents the evaluation of the observable sensitivities $\partial y_i / \partial \theta_k$ by introducing the adjoint state $\mathbf{p}(t, \boldsymbol{\theta}, \mathbf{u}) : [t_0, t_{n_t}] \times \mathbb{R}^{n_\theta} \times \mathbb{R}^{n_u} \to \mathbb{R}^{n_x}$, such that $\forall j = n_t \ldots, 1$ on interval $(t_{j-1}, t_j]$ it satisfies the backward differential equation

$$\dot{\mathbf{p}}(t, \boldsymbol{\theta}, \mathbf{u}) = -\mathbf{J}(\mathbf{x}(t, \boldsymbol{\theta}, \mathbf{u}), \boldsymbol{\theta}, \mathbf{u})^T \mathbf{p}(t, \boldsymbol{\theta}, \mathbf{u}), \tag{9}$$

with boundary values

$$\mathbf{p}(t_j, \boldsymbol{\theta}, \mathbf{u}) = \lim_{t \to t_j^+} \mathbf{p}(t, \boldsymbol{\theta}, \mathbf{u}) + \sum_{j=1}^{n_y} \left. \frac{\partial h_i}{\partial \mathbf{x}} \right|_{(\mathbf{x}(t_j, \boldsymbol{\theta}, \mathbf{u}), \boldsymbol{\theta}, \mathbf{u})}^T \frac{(\bar{y}_{ij} - y_i(t_j, \boldsymbol{\theta}, \mathbf{u}))}{\sigma_{ij}^2}, \quad \text{and}$$

$$\lim_{t \to t_{n_t}^+} \mathbf{p}(t, \boldsymbol{\theta}, \mathbf{u}) = 0,$$

and Jacobian matrix $\mathbf{J}$,

$$\mathbf{J}(\mathbf{x}(t, \boldsymbol{\theta}, \mathbf{u}), \boldsymbol{\theta}, \mathbf{u}) = \begin{bmatrix} \dfrac{\partial f_1}{\partial x_1} & \cdots & \dfrac{\partial f_1}{\partial x_{n_x}} \\ \vdots & \ddots & \vdots \\ \dfrac{\partial f_{n_x}}{\partial x_1} & \cdots & \dfrac{\partial f_{n_x}}{\partial x_{n_x}} \end{bmatrix}_{\mathbf{x}(t, \boldsymbol{\theta}, \mathbf{u}), \boldsymbol{\theta}, \mathbf{u}}. \tag{10}$$

Given the adjoint state, the objective function gradient can be computed as

$$\left. \frac{\partial \mathcal{J}}{\partial \theta_k} \right|_{\boldsymbol{\theta}} = -\sum_{i=1}^{n_y} \sum_{j=1}^{n_t} \frac{(\bar{y}_{ij} - y_i(t_j, \boldsymbol{\theta}, \mathbf{u}))}{\sigma_{ij}^2} \left. \frac{\partial h_i}{\partial \theta_k} \right|_{\mathbf{x}(t, \boldsymbol{\theta}, \mathbf{u}), \boldsymbol{\theta}, \mathbf{u}}$$

$$- \int_{t_0}^{t_{n_t}} \mathbf{p}(t, \boldsymbol{\theta}, \mathbf{u})^T \left. \frac{\partial \mathbf{f}}{\partial \theta_k} \right|_{\mathbf{x}(t, \boldsymbol{\theta}, \mathbf{u}), \boldsymbol{\theta}, \mathbf{u}} dt - \mathbf{p}(t_0, \boldsymbol{\theta}, \mathbf{u})^T \left. \frac{\partial \mathbf{x}_0}{\partial \theta_k} \right|_{\boldsymbol{\theta}, \mathbf{u}}, \tag{11}$$

in which $\partial \mathbf{x}_0 / \partial \theta_k$ denotes the sensitivity of the initial state with respect to parameter $\theta_k$. If the initial condition is given as an explicit function, this derivative can be computed easily; however, its calculation is non-trivial in the case of pre-equilibration.

To use ASA for the pre- and post-equilibration cases, the steady-state calculation is currently performed via a long simulation (see, e.g. [12, 27]):

1. For the case of a time-course followed by a post-equilibration, the simulation is run until a time $t'' \gg t_{n_t}$ for which time derivatives $\dot{\mathbf{x}}$ are negligible. We used (3) as convergence criterion. The values of the observables at time $t''$, $y_i(t'', \boldsymbol{\theta}, \mathbf{u})$, are considered a good approximation of the observables at the steady state, $y_{i^*}(\boldsymbol{\theta}, \mathbf{u})$, and used for the evaluation of the objective function.

2. For the case of pre-equilibration followed by a time-course, in a first step a simulation is run for the pre-equilibration condition ($\mathbf{u}^e$) until the state derivatives $\dot{\mathbf{x}}$ are sufficiently small. The initial time point for this simulation is denoted by $-t'$. In the second step, the

condition is switched to **u** and a simulation is performed for the transient phase using the final state of the pre-equilibration simulation as the initial state. This can be interpreted as running the simulation for $t \in [-t', t_0]$ under the pre-equilibration condition ($\mathbf{u}^e$) and then switching for $t \in (t_0, t_{n_t}]$ to the time course condition (**u**).

These approximations of the exact setup are good if the convergence criteria are tight. In practice, one often uses values on the order of $10^{-8}$ and $10^{-16}$ as the relative and absolute tolerance, respectively [31]. One should note, that ASA only provides the objective function gradient, but not the sensitivity of the steady state with respect to the parameters.

## Adjoint sensitivity analysis at steady state (ssASA)

As steady-state constraints are encountered in a large fraction of application problems (see, e.g., the benchmark collection in [28]), we propose here an adjoint sensitivity analysis method that is tailored for problems with pre- and post-equilibration.

**Post-equilibration case.**   For experiments with a post-equilibration, steady-state measurements ($\mathcal{D}_*$) are available in addition to time-course measurements ($\mathcal{D}_t$). To compute the objective function gradient (11), one needs to compute the term

$$\int_{t_0}^{t''} \mathbf{p}(t, \boldsymbol{\theta}, \mathbf{u})^T \frac{\partial \mathbf{f}}{\partial \theta_k}\bigg|_{\mathbf{x}(t, \boldsymbol{\theta}, \mathbf{u}), \boldsymbol{\theta}, \mathbf{u}} dt. \tag{12}$$

Note, that the upper limit of the integral is different from the one in (11) as the steady-state measurements time point $t = t''$ were added. On the time interval $[t_0, t_{n_t})$ the integral should be computed using the standard ASA, while on the interval $[t_{n_t}, t'']$ the proposed approach can be applied.

Let us introduce a time point $t'$ such that $t_{n_t} \ll t' \ll t''$ and for $t \geq t'$ the system (1) is at steady state. Then for $t \geq t'$ the Jacobian (10) is a constant matrix. Due to the exponential stability of the steady state, the precise choice of $t'$ is not important, as long as $t'$ has been chosen large enough. Then, on the time interval $t \geq t'$, the adjoint system of ODEs (9) simplifies to

$$\dot{\mathbf{p}}(t, \boldsymbol{\theta}, \mathbf{u}) = -\mathbf{J}(\mathbf{x}^*(\boldsymbol{\theta}, \mathbf{u}), \boldsymbol{\theta}, \mathbf{u})^T \mathbf{p}(t, \boldsymbol{\theta}, \mathbf{u}). \tag{13}$$

This system has the solution

$$\mathbf{p}(t, \boldsymbol{\theta}, \mathbf{u}) = e^{-\mathbf{J}(\mathbf{x}^*(\boldsymbol{\theta}, \mathbf{u}), \boldsymbol{\theta}, \mathbf{u})^T (t - t'')} \mathbf{p}(t'', \boldsymbol{\theta}, \mathbf{u}).$$

The steady state of the system (1) is exponentially stable if and only if the eigenvalues of the Jacobian at the steady state ($\mathbf{J}(\mathbf{x}^*(\boldsymbol{\theta}, \mathbf{u}), \boldsymbol{\theta}, \mathbf{u})$) have negative real parts. This follows that the steady state $\mathbf{p} = \mathbf{0}$ of the system (13) is asymptotically stable in reverse time. Hence, as the time interval $[t', t'']$ can be chosen large enough, it holds that

$$\lim_{t \to t'^+} \mathbf{p}(t, \boldsymbol{\theta}, \mathbf{u}) = \mathbf{0},$$

which is illustrated in Fig 2a. As the steady state $\mathbf{p} = \mathbf{0}$ is stable, the system (13) will not diverge from it on the interval $[t_{n_t}, t']$.

As a consequence, the integral

$$\int_{t_{n_t}}^{t''} \mathbf{p}(t, \boldsymbol{\theta}, \mathbf{u})^T \frac{\partial \mathbf{f}}{\partial \theta_k}\bigg|_{\mathbf{x}(t, \boldsymbol{\theta}, \mathbf{u}), \boldsymbol{\theta}, \mathbf{u}} dt. \tag{14}$$

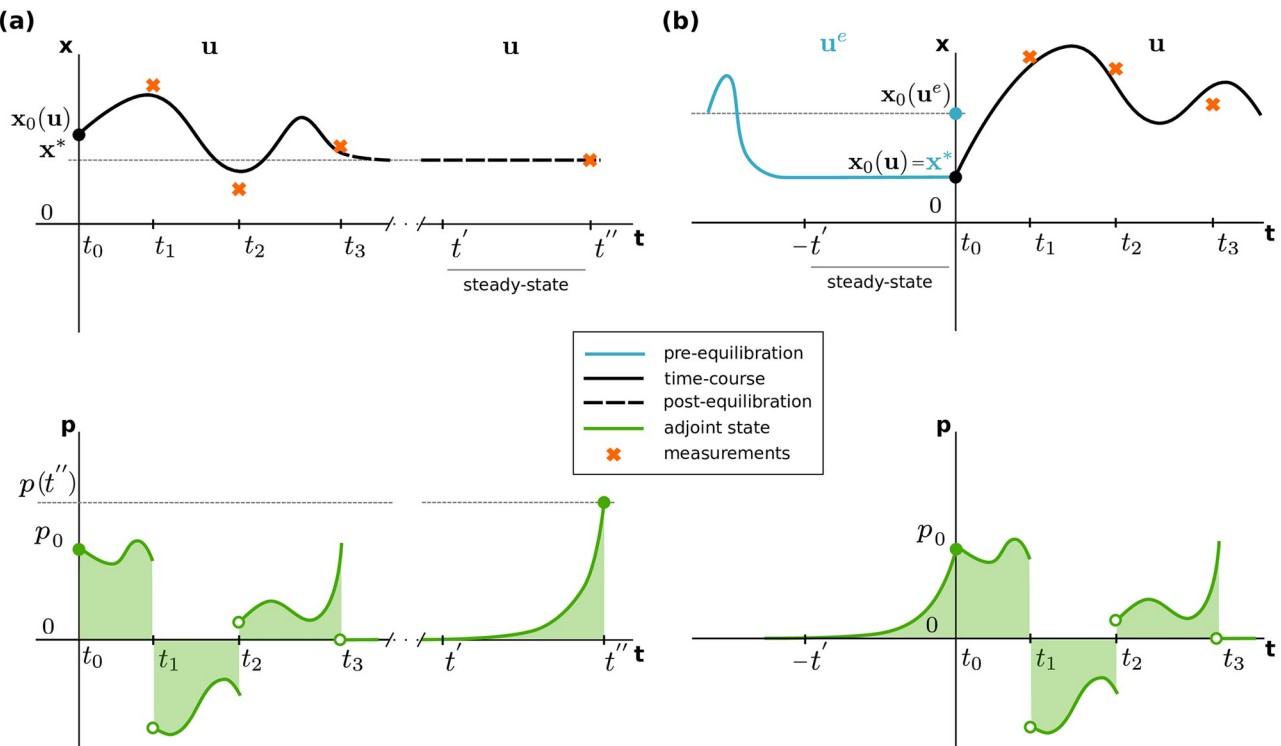

**Fig 2. Computation of the adjoint state integral term.** The top row illustrates the solution of the system (1) for (a) the post-equilibration case and (b) the pre-equilibration case for $n_t = 3$. The bottom row illustrates the integral term in (11).

reduces to a matrix-vector product:

$$\int_{t'}^{t''} \mathbf{p}(t, \boldsymbol{\theta}, \mathbf{u})^T dt \cdot \frac{\partial \mathbf{f}}{\partial \theta_k}\bigg|_{\mathbf{x}=\mathbf{x}^*(\boldsymbol{\theta}, \mathbf{u})} dt = \mathbf{p}_{\text{integral}} \cdot \frac{\partial \mathbf{f}}{\partial \theta_k}\bigg|_{\mathbf{x}=\mathbf{x}^*(\boldsymbol{\theta}, \mathbf{u})},$$

since

$$\int_{t_{n_t}}^{t'} \mathbf{p}(t, \boldsymbol{\theta}, \mathbf{u})^T dt \cdot \frac{\partial \mathbf{f}}{\partial \theta_k}\bigg|_{\mathbf{x}=\mathbf{x}^*(\boldsymbol{\theta}, \mathbf{u})} dt = 0.$$

To obtain the integral one can solve the linear system of equations

$$\mathbf{J}(\mathbf{x}^*(\boldsymbol{\theta}, \mathbf{u}), \boldsymbol{\theta}, \mathbf{u})^T \mathbf{p}_{\text{integral}} = -\mathbf{p}(t'', \boldsymbol{\theta}, \mathbf{u}), \tag{15}$$

see Section "Adjoint state integral computation at steady state" of the S1 Text for the derivation. Additionally, the calculations from this section performed for a simple example of a conversion reaction can be found in Section "Conversion reaction example" of the S1 Text.

**Pre-equilibration case.** For experiments with a pre-equilibration, in order to do standard ASA, one requires initial state sensitivities ($\frac{\partial \mathbf{x}_0}{\partial \theta_k}$) (see Eq (11)). They can be computed by performing forward sensitivity analysis during pre-equilibration.

Alternatively, instead of computing $\frac{\partial \mathbf{x}_0}{\partial \theta_k}$, the ASA method can be extended to the pre-equilibration time interval $[-t', t_0]$. On the intervals $[t_N, t_{N-1}), \ldots, [t_1, t_0)$ standard ASA can be performed as described in Section "Adjoint sensitivity analysis (ASA)". As described for the post-

equilibration case, at $t = -t'$, the system (13) is at steady state $\mathbf{p} = \mathbf{0}$, hence the scalar product of the initial state sensitivities of pre-equilibration, with the adjoint state at $t = -t'$, vanishes. Therefore, one only needs to compute the quadratures

$$\int_{-t'}^{t_0} \mathbf{p}(t, \boldsymbol{\theta}, \mathbf{u}^e)^T \left. \frac{\partial \mathbf{f}}{\partial \theta_k} \right|_{\mathbf{u}=\mathbf{u}^e, \mathbf{x}=\mathbf{x}_0} dt. \tag{16}$$

For this, the adjoint state from the time-course adjoint simulation must be passed on to the initial adjoint state of backward pre-equilibration ($\mathbf{p}(t_0, \boldsymbol{\theta}, \mathbf{u})$). Accordingly, for the case of pre-equilibration, the ODEs solved for pre-equilibration and time-dependent observations are coupled, and cannot be computed independently.

Similarly to the post-equilibration case, there are two possibilities for computing this integral. The first possibility is to compute it by simulating the system (1) until it reaches its steady state and then compute the integral (16) backward on the same time interval. The second possibility is to use the proposed ssASA method. One can assume, without loss of generality, that the system (1) is at the steady state that corresponds to the pre-equilibration condition for all $t < t_0$. On this interval, the same considerations as in the previous section apply and the adjoint system simplifies to (13), but in this case $\mathbf{x}^*(\boldsymbol{\theta}, \mathbf{u}) = \mathbf{x}^*(\boldsymbol{\theta}, \mathbf{u}^e)$. To find the integral one can solve the linear system of equations

$$\mathbf{J}(\mathbf{x}^*(\boldsymbol{\theta}, \mathbf{u}^e), \boldsymbol{\theta}, \mathbf{u}^e)^T \mathbf{p}_{\text{integral}} = -\mathbf{p}(t_0, \boldsymbol{\theta}, \mathbf{u}). \tag{17}$$

This case is illustrated in Fig 2b.

**Applicability of the proposed method.** The systems (15) and (17) have a unique solution only if the (transposed) Jacobian is non-singular. Otherwise, the proposed approach is not applicable. A common cause for singular Jacobians is, amongst others (cf. [32] for a review), presence of conserved quantities in the system. Conserved quantities or conserved moieties are functions of the sates of the dynamical system that remain constant over time. Identifying each conserved quantity allows to reduce model dimension by excluding one state, expressing it in terms of other states contained in the conserved quantity. A simple example can be found in Section "Conversion reaction example" of the S1 Text. There are various deterministic [33, 34] algorithms available to identify conserved quantities, which, however, are not suitable for large/genome-scale reaction networks due to combinatorial complexity. A fast, scalable, heuristic-based approach presented in [35] is better suited for large-scale reaction networks.

Removing conserved quantities from the system allows to apply the proposed method. If it is not possible or if the Jacobian remains singular, the adjoint state ODEs have to be integrated numerically.

## Implementation

The new algebraic approach for computing objective function gradient as well as a heuristic-based conserved moieties identification approach [35] have been made available via the open-source AMICI package (https://github.com/AMICI-dev/AMICI/) [36], which interfaces the SUNDIALS solver CVODES [37]. Implementation details can be found in Section "AMICI Implementation" of the S1 Text.

During the numerical study, described in the next section, the AMICI package was used for model simulations [38]. Parameter estimation was performed using the open-source Python Parameter EStimation TOolbox (pyPESTO) [39], which internally used the Interior Trust Region Reflective algorithm for boundary constrained optimization problems implemented in the Fides Python package for optimization [40, 41].

## Results

In this section we investigate whether using the proposed ssASA method is beneficial, compared to the standard ASA approach, to computing the objective function gradient. In both cases we used ASA, but in the first case gradient computation was done via numerical integration of the ODE system (9), whereas in the second case the new approach was applied, where the system of Eqs (15) and (17) is solved. In both cases numerical integration was used to find the steady state of the system (1), i.e. to find the solution $\mathbf{x}^*(\boldsymbol{\theta}, \mathbf{u})$ of $\dot{\mathbf{x}} = \mathbf{f}(\mathbf{x}^*(\boldsymbol{\theta}, \mathbf{u}), \boldsymbol{\theta}, \mathbf{u}) = \mathbf{0}$. The two cases will be referred to as "standard ASA for sensitivities" and "ssASA for sensitivities". We assess accuracy by comparing the gradient values computed with the two methods, as well as the difference in computation time.

The comparison is done on three previously published ODE-constrained optimization problems of varying complexity that required either pre- or post-equilibration (Table 1). The problems were taken from the PEtab Benchmark Collection, a GitHub repository with parameter estimation problems from published studies implemented in the PEtab format [11, 42].

The Blasi *et al.*, 2016 model describes acetylation of the histone H4 [13]. It is linear and has 16 state variables and 9 optimized parameters, which is the dimension of objective function gradient (8). Only steady-state data is available for this model. The Blasi *et al.*, 2016 model contains conserved quantities, which were automatically removed during model import by AMICI to ensure non-singularity of the Jacobian. The model by Zheng *et al.*, 2012 is also linear and describes histone H3 methylation pattern formation. For this model dynamic data is available and the initial condition is determined by pre-equilibration. The model by Fröhlich *et al.*, 2018 is a large-scale nonlinear model describing the drug response of cancer cell lines [12]. The measurement data used for optimization was collected when the system was assumed to be at steady state.

### Accuracy with ssASA is preserved

To investigate the accuracy of objective function gradients computed with the proposed ssASA method, we performed 500 model simulations with different parameter vectors sampled uniformly within the parameter bounds specified in the parameters PEtab file [11]. For each model, the same initial state, specified in the PEtab files, was used for each simulation and no simulation failed.

For all three models the computed objective function gradients were very similar for both compared approaches (Fig 3). The maximum and median deviations (as defined in the S1 Text, Section "Accuracy of gradient computation") are respectively equal to $1.9 \cdot 10^{-4}$ and numerically 0 for the Blasi *et al.*, 2016 model, $2.8 \cdot 10^{-11}$ and $1.6 \cdot 10^{-17}$ for the Zheng *et al.*, 2012 model, $3.1 \cdot 10^{-2}$ and $3.04 \cdot 10^{-20}$ for the Fröhlich *et al.*, 2018 model. Therefore, we conclude that the new approach gives accurate results.

**Table 1. Overview of the models and optimization problems considered in this study.**

| Problem | $n_\theta$ | $n_x$ | # data points | Equilibration type | Reference |
|---|---|---|---|---|---|
| Blasi *et al.*, 2016 | 9 | 16 | 288 | post-equilibration | [13] |
| Zheng *et al.*, 2012 | 46 | 15 | 60 | pre-equilibration | [14] |
| Fröhlich *et al.*, 2018 | 4088 | 1396 | 143 | post-equilibration | [12] |

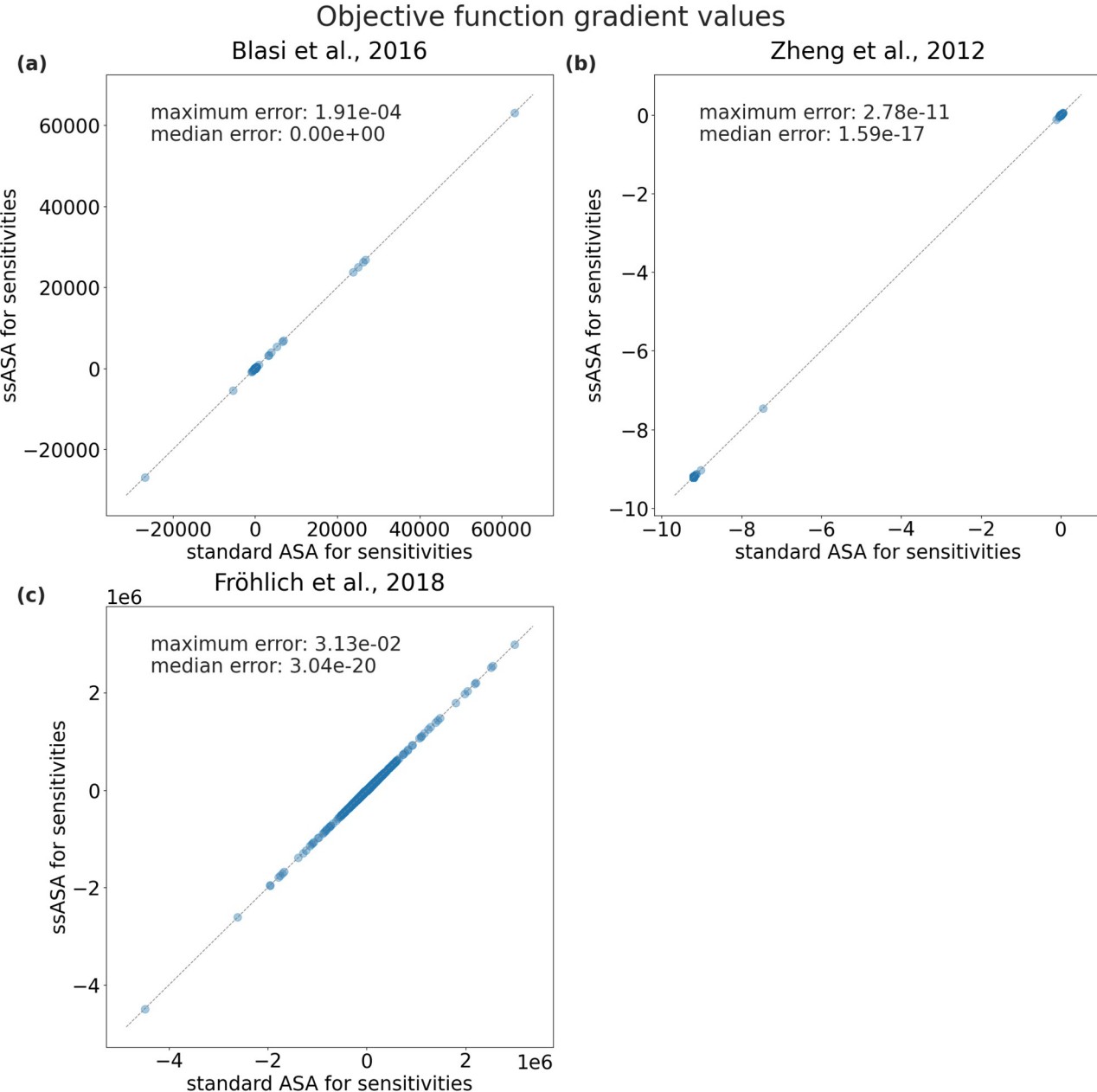

**Fig 3. Accuracy of objective function gradients of the proposed method.** In each of the three subplots one point corresponds to an objective function gradient value computed during one simulation using either standard ASA (x-axis) or ssASA (y-axis) for sensitivities computation. The number of points in each subplot is equal to the number of optimized parameters multiplied by the number of simulations. Points close to the diagonal indicate a good agreement.

## Simulation with ssASA is faster

We then compared how simulation time differs for the two approaches. For all three models, all simulations were significantly faster with the new approach (Fig 4). The precise speedup depends on the model. The simulations were on average 3.3, 1.2, 3.3 times faster with the new

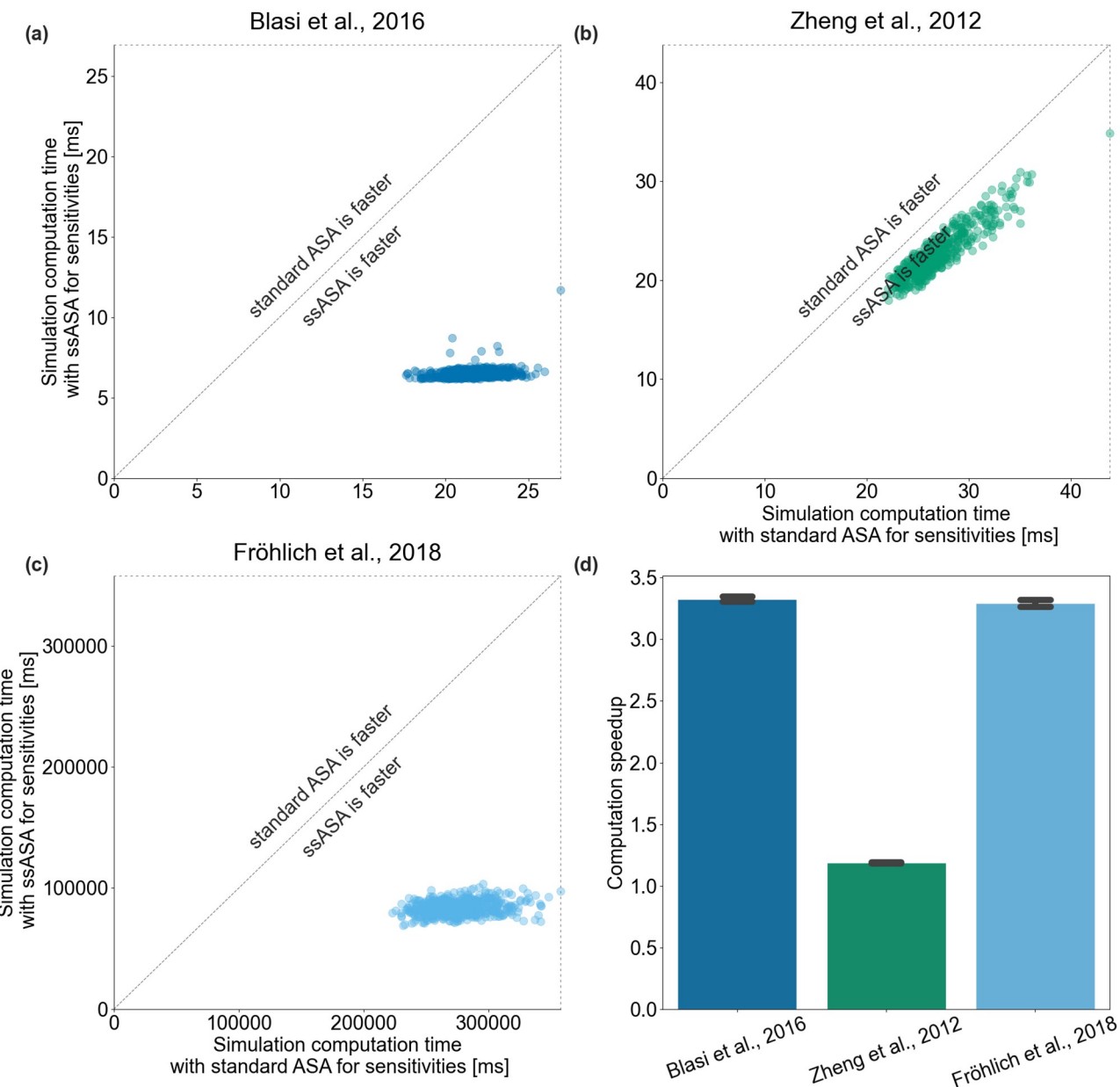

**Fig 4. Simulation efficiency of the proposed method.** Each point corresponds to a total simulation time with ASA (x-axis) and ssASA (y-axis) for sensitivities computation. Points on the diagonal correspond to simulations that took equal time with both approaches. (a) Blasi *et al.*, 2016, (b) Zheng *et al.*, 2012 and (c) Fröhlich *et al.*, 2018 models. (d) Computation speedup of simulations using ssASA for sensitivities computation compared to using standard ASA. Each bar height corresponds to a mean of computation speedups of all simulations and each error bar corresponds to the sample standard deviation.

approach for the Blasi *et al.*, 2016, Zheng *et al.*, 2012 and Fröhlich *et al.*, 2018 models, respectively (Fig 4d).

Additionally, we investigated how simulation time and computation speedup scale with respect to the number of optimized parameters. We simulated the Fröhlich *et al.*, 2018 model with 11 different estimation problem sizes (1, 2, 5, 12, 28, 64, 147, 337, 775, 1780, or 4088 estimated parameters), each 200 times, for a total of 2200 simulations. For each simulation, the

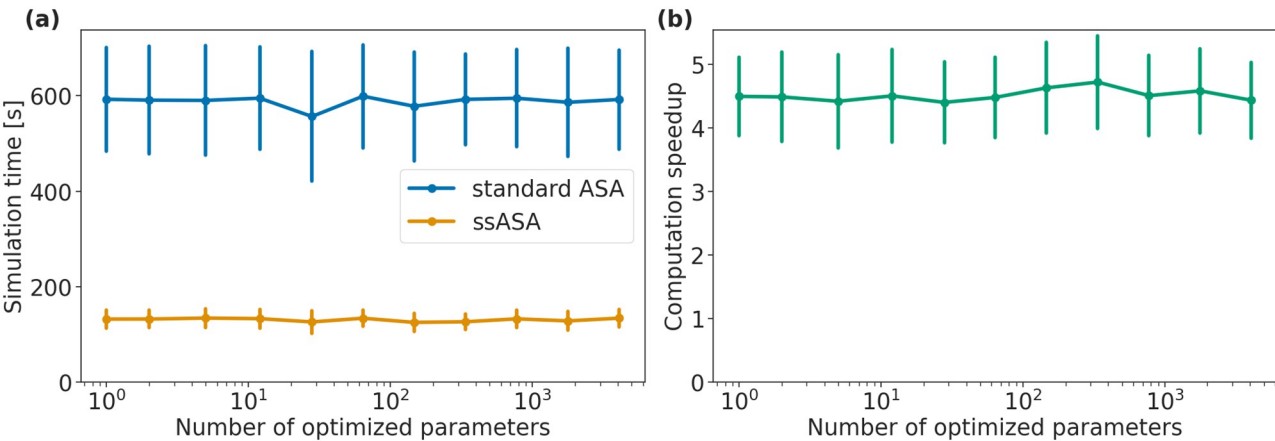

**Fig 5. Scaling of simulation time and computation speedup of simulations with respect to the number of optimized parameters.** Fröhlich *et al.*, 2018 model, 200 simulations with 1, 2, 5, 12, 28, 64, 147, 337, 775, 1780, 4088 optimized parameters. (a) Each point is the mean simulation time for corresponding number of free parameters, and the error bars are the standard deviations. (b) Each point represents computation speedup of simulations performed with ssASA for sensitivities compared to using standard ASA for sensitivities, the error bars show the standard deviation.

subset of parameters set to be estimated or fixed was randomly chosen, and all parameter values were sampled uniformly within the parameter bounds in the PEtab parameters table. During simulation, sensitivities of the objective function were only computed for the estimated parameters. As expected, simulation time did not change with the number of optimized parameters for both sensitivities computation methods, (Fig 5a), as the dimension of both the adjoint state ODE (9) and the system (15) do not change with the number of optimized parameters and is equal to $n_x$. Consequently, computation speedup does not change with the number of optimized parameters, and the proposed method is faster, independent of the number of free parameters (Fig 5b).

## Optimization with ssASA is faster

Furthermore, we investigated how the speedup we observed for individual model simulations with different parameter vectors translates to parameter estimation tasks. A large number of model simulations might be required during optimization for parameter estimation, which also means that if pre- or post-equilibration is needed it will be executed during each simulation.

We compared the difference in computation time for solving parameter estimation problems using multi-start local optimization between the two approaches for sensitivities computation, (Fig 6a–6c). Same as for the simulation comparison, initial parameter vectors were sampled uniformly within the parameter bounds specified in the parameters PEtab file. The same initial parameter values were used for optimizations with ASA and ssASA. For the Blasi *et al.*, 2016 and Zheng *et al.*, 2012 models we ran 500 multi-starts. As the Fröhlich *et al.*, 2018 model is computationally demanding, in this study, we considered only a subset of data used in [12]. We used only the control conditions, that is 143 out of the total 5281 conditions. For this model, we ran 50 multi-start optimizations with the number of optimization steps limited to 50. This would not yield good model fits, but should provide a representative performance estimate.

For the Blasi *et al.*, 2016 and Zheng *et al.*, 2012 models all multi-starts finished successfully. For the Fröhlich *et al.*, 2018 model for both methods one out of 50 multi-starts failed at the

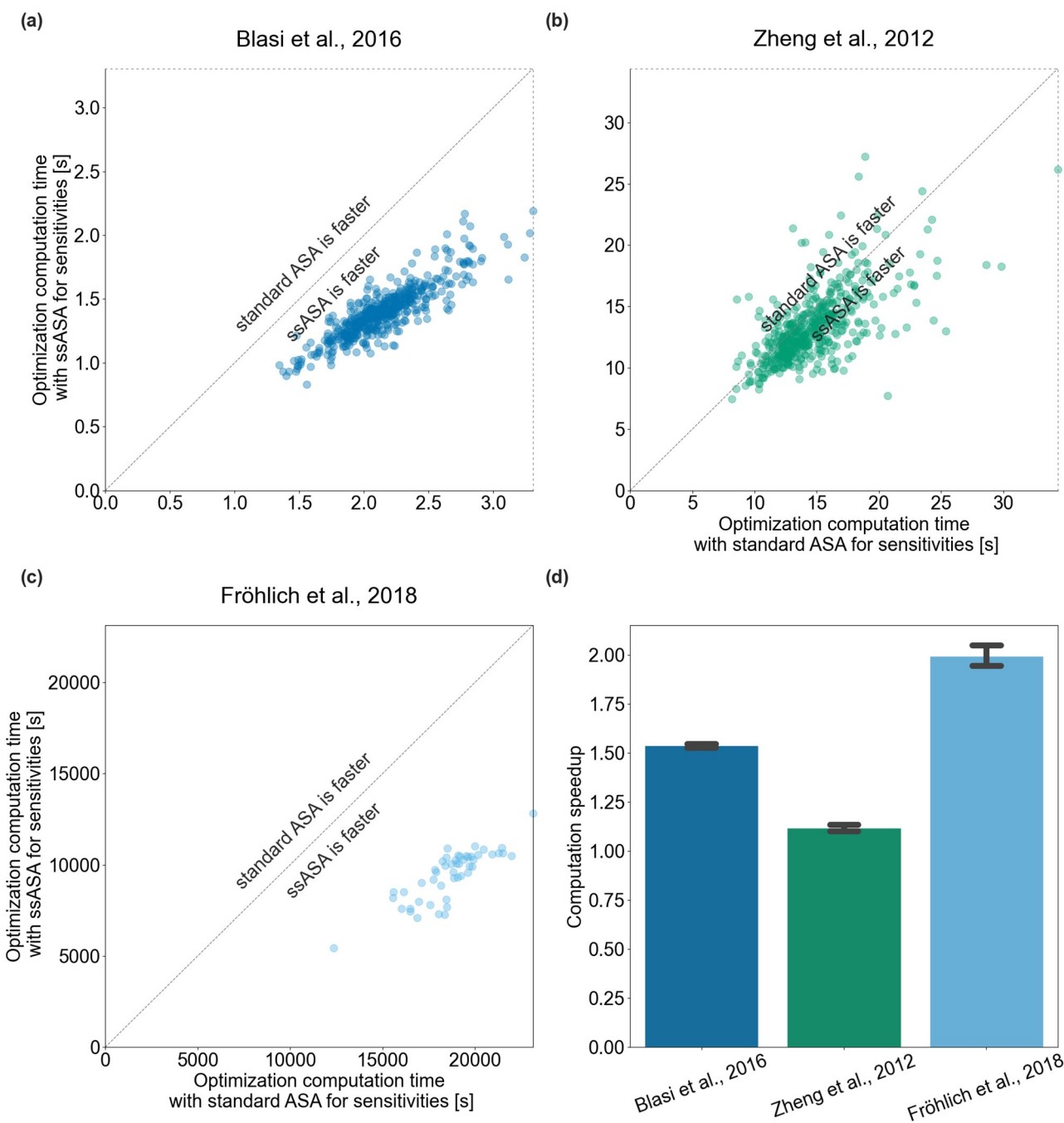

**Fig 6. Optimization efficiency of the proposed method.** Each scatter point shows the total computation time required for one multi-start optimization using standard ASA (x-axis) or ssASA (y-axis) for sensitivities computation (a) Blasi *et al.*, 2016 model, (b) Zheng *et al.*, 2012 model, (c) Fröhlich *et al.*, 2018 model. Points on the diagonal correspond to multi-starts that took equal time with both approaches. (d) Computation speedup of optimizations using ssASA for sensitivities computation compared to using standard ASA for sensitivities. Each bar height corresponds to a mean of multi-start local optimization computation speedups and each error bar corresponds to the sample standard deviation.

first step and the other 49 finished as the number of optimization steps exceeded 50. Optimizations were on average 1.5, 1.1, 1.9 times faster with the new approach for Blasi *et al.*, 2016, Zheng *et al.*, 2012 and Fröhlich *et al.*, 2018 models respectively (Fig 6d).

The speedups for optimization were lower than those observed for model simulation, because the proposed method only affects simulation time, but not the time taken in the optimizer itself, or other overhead. The Blasi *et al.*, 2016 model is rather small and linear and simulations made up only approximately 50% of total optimization time. Therefore, the 3-fold speedup in simulations resulted in 1.3-fold speedup of optimizations. For the Zheng *et al.*, 2012 model speedup is about the same as for simulations. For the Fröhlich *et al.*, 2018 model simulations made up around 25% of the total optimization time. Here, model simulation is very costly, but also significant time was spent in the optimizer due to the large number of optimized parameters. Generally, the speedup for optimization will be highly implementation-dependent.

## Discussion and conclusion

It has been shown, that if finding a steady state is required during model simulation, whether it is pre-equilibration, post-equilibration, or both, it is possible to partially avoid numerical integration and to solve a system of algebraic equations instead [21]. In this study, we showed that this idea can also be used to simplify objective function gradient computation. We introduced a new method for efficient adjoint sensitivities computation at steady state. We showed that an intermediate step of objective function gradient computation with ASA, i.e., backward numerical integration of the adjoint state ODEs, can be avoided by solving a linear system of equations for the time interval including steady-state measurement time point. We also showed that in case pre-equilibration is required, computation of initial state sensitivities $(\frac{\partial \mathbf{x}_0}{\partial \theta_k})$ with forward sensitivity analyses can be avoided by extending ASA to the pre-equilibration interval. The proposed method is applicable on this time interval as well, and a linear system of equations can be solved instead of backward numerical integration.

We described how the new approach can be applied in the case of modelling biochemical processes, tested it on three published models and demonstrated that it is accurate and allows for efficient sensitivity computation. The exact speedup over the standard approach depends on the model and dataset. More specifically, it depends on the computational complexity of computing the integral (14) (or (16) for the pre-equilibration case) on the time interval between the last finite time point ($t_{n_t}$) and the time point where the steady-state condition is satisfied. It is the difference between solving an $n_x$-dimensional linear non-autonomous ODE system, and solving a linear system of $n_x$ equations. The speedup is expected to be higher for nonlinear systems. It also depends on the fraction of computational cost spent on computing the steady-state. For example, if the time spent in the optimizer is large compared to the time for model simulation, or if mostly time-course data is available, the speedup will be lower. As expected, the speedup does not significantly depend on the number of optimized parameters, which we demonstrated with the example of one of the three models. We observed a speedup of up to 3.3-fold for simulations and up to two-fold for parameter estimation, for the models and datasets considered in this study.

The new approach is only applicable if for each initial value there exists a unique, exponentially stable steady state. This implies that the (transposed) Jacobian of the right-hand side is non-singular. One of the main reasons for a singular Jacobian is the presence of conserved quantities in the dynamical system [32]. In order to facilitate applicability of the proposed method we have implemented a fast heuristic-based approach for conserved quantities identification suitable for large-scale reaction networks as presented in [35]. Both the conserved moieties identification approach and the proposed ssASA approach have been made available via the open-source AMICI package. If the uniqueness of the steady state cannot be ensured, e.g. because the system might be multi-stable, the method has to be applied carefully. While

there also is a unique mapping from initial condition to steady state for multi-stable systems with Lipschitz-continuous right-hand sides, this mapping might not be obeyed by Newton's method. We do not observe this problem in practice, but recommend to check this after optimization and to provide Newton's method with a refined initial value (e.g. obtained via a short simulation run, which is already supported by AMICI) if necessary. We have tested this case on the Blasi et al., 2016 model. Conserved quantities have been removed before applying the proposed ssASA approach.

Solving parameter estimation problems consist of various steps that depend on the model and available data. Different methods, for finding steady states, numerical integration, or sensitivity computation, might be applicable or more efficient for a particular model. Further analysis and benchmarking of different approaches and their combinations would be beneficial. For example, a broader study of the steady-state case that considers different methods for steady-state computation in combination with different approaches for sensitivity analysis for both forward and adjoint sensitivity methods, to find especially efficient combinations, would be of interest.

## Supporting information

**S1 Text. Supplementary notes on the ssASA method derivation, accuracy assessment and AMICI implementation, as well as the conversion reaction example.**
(PDF)

## Author Contributions

**Conceptualization:** Polina Lakrisenko, Paul Stapor, Stephan Grein, Dilan Pathirana, Daniel Weindl, Jan Hasenauer.

**Formal analysis:** Polina Lakrisenko.

**Funding acquisition:** Glenn Terje Lines, Daniel Weindl, Jan Hasenauer.

**Investigation:** Polina Lakrisenko.

**Methodology:** Paul Stapor, Stephan Grein, Dilan Pathirana, Daniel Weindl, Jan Hasenauer.

**Software:** Polina Lakrisenko, Paul Stapor, Stephan Grein, Łukasz Paszkowski, Dilan Pathirana, Fabian Fröhlich, Daniel Weindl.

**Supervision:** Dilan Pathirana, Glenn Terje Lines, Daniel Weindl.

**Visualization:** Polina Lakrisenko.

**Writing – original draft:** Polina Lakrisenko.

**Writing – review & editing:** Polina Lakrisenko, Paul Stapor, Stephan Grein, Łukasz Paszkowski, Dilan Pathirana, Fabian Fröhlich, Glenn Terje Lines, Daniel Weindl, Jan Hasenauer.

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
