## [Decision Letter · Decision Letter 0]

11 Sep 2022

Dear Lakrisenko,

Thank you very much for submitting your manuscript "Efficient computation of adjoint sensitivities at steady-state in ODE models of biochemical reaction networks" for consideration at PLOS Computational Biology.

As with all papers reviewed by the journal, your manuscript was reviewed by members of the editorial board and by several independent reviewers. In light of the reviews (below this email), we would like to invite the resubmission of a significantly-revised version that takes into account the reviewers' comments.

Please take into account the comments of the three referees, especially highlighting the new theoretical aspects of the work, in lght of the comments of Reviewer 3 on Green finctions.

We cannot make any decision about publication until we have seen the revised manuscript and your response to the reviewers' comments. Your revised manuscript is also likely to be sent to reviewers for further evaluation.

Sincerely,

Attila Csikász-Nagy

Academic Editor

PLOS Computational Biology

Mark Alber

Section Editor

PLOS Computational Biology

Reviewer's Responses to Questions

**Comments to the Authors:**

Reviewer #1: The paper by Lakriscnko et al presents a new numerical algorithm for sensitivity analysis of ODE models, mostly applied to biochemical reaction networks. They focus on in particular speeding up situations where steady-state data are available and of interest, in addition to dynamic data as typically. Overall the manuscript is well written, easy to follow, and presents a significant advance in this class of problems, particularly for larger models where the differences of speed are important, and which are becoming more and more prevalent. I only have some minor comments that the authors may consider in a revision. First, I’d like to have seen a little more analysis or speculation / discussion as to why speed up is bigger for some models and not others, and how this speed up scales with problem sizes. Fig. 4 and 6 show some of this data but one is left wondering if there is any insight for future models, what the speed up may be based on certain properties. A key weakness was inapplicability to models with conserved quantities (singular Jacobian). It would have also been nice to see the implementation on a model that had a singular Jacobian due to conserved quantities, but the tools being mentioned as solutions to this caveat (e.g. Ref 35) were applied and then the presented approach was successfully employed.

Reviewer #2: This paper proposes a useful approach for the sensitivity computation of dynamical models taking into consideration steady state data. It is an appealing property of the method that the computation burden does not depend on the number of parameters. The advantages of the approach over existing popular methods are well demonstrated on the case studies taken from the literature. The supporting information is also useful for better understanding. My main overall comment is that the assumptions, notations and applicability conditions should be made more precise in the paper before publication.

Specific comments:

1) The title contains 'biochemical reaction networks' although the proposed method does not seem to be specific to reaction networks or even to nonnegative biological models. Please clarify the system class to which the methodology is applicable.

2) The notation in eq. (1) is odd: if we want to denote all arguments correctly, I think

$\\dot{x} = f(x(t), \\theta, u(t))$ would be sufficient provided that the input is time-varying and the parameters are assumed to be constant. This makes $x$ to depend on $\\theta$ and $u$ since the derivative depends on them. Moreover:

- It is not clear whether the input is constant or time-varying. Can it be considered as a vector of known parameters?

- Why does the initial condition directly depend on the parameters? This is also unusual.

3) Are there any assumptions on the allowed state and parameter dependence of the dynamical model? (e.g., (quasi)polynomial, rational etc.)

4) Please comment on the applicability of the approach for models with steady state multiplicity, since these are quite common among biological systems.

5) In Fig 1.a, the equilibrium $x^*$ depends on $u$. This is suspicious. (Why not on $u^e$?)

6) On page 5 the authors write:

"The results also apply to unknown noise variances and other noise distributions."

I think that at least the mean of the noise should be zero and the noise components should be uncorrelated at different time instants, otherwise the parameter estimation will be biased.

Reviewer #3: It is well known that inhomogeneous linear ODEs can be solved in two ways: (1) solving the equations directly; (2) solving the corresponding homogeneous linear ODE. The solution is called the adjoint function or Green's function. Then, the Green function is integrated with the inhomogeneous part, providing the solution of the original ODE.

George Green (14 July 1793–31 May 1841) was a British mathematician and physicist, who wrote "An Essay on the Applications of Mathematical Analysis to the Theories of Electricity and Magnetism" (Green, 1828). The essay introduced several important concepts, among them a theorem similar to modern Green's theorem, the idea of potential functions as currently used in physics, and the concept of what are now called Green's functions.

https://www.nottingham.ac.uk/physics/about/history/george-green.aspx

Green published this idea in 1828, therefore the following sentence of the authors is not well founded:

"This approach has been developed in [Plessix RE, Geophysical Journal International. 2006]. The first application of the adjoint function method (also called Green's Function Method, GFM) in chemical kinetics for the calculation of sensitivity coefficients were published by J. Hwang et al. (The Green's function method of sensitivity analysis in chemical kinetics, The Journal of Chemical Physics 69 (1978) 5180-5191.)

I recommend the authors a recent paper about the calculation of sensitivity coefficients in chemical kinetics using the GFM: M. Lemke et al. Combustion Theory and Modelling, 23, 180-196 (2019),

https://doi.org/10.1080/13647830.2018.1495845 ) In this paper, references [16]-[27] discuss the history of the GFM in the analysis of chemical kinetic models.

The authors do not solve the differential equations, but calculate the sensitivity coefficients in the stationary point, when the ODEs are reduced to linear algebraic equations. These matrix equations were derived from the GFM concept. This is also an obvious step. The paper does not contain any new math and very little novelty from a computational point of view. However, I still consider the publication of the paper possible, since the authors well justify that the software they created and made publicly available can be useful for the interpretation and optimization of some systems biology measurements. Surely, selected papers from references [16]-[27] have to be added to the manuscript.

"The new approach is only applicable if the (transposed) Jacobian is non-singular.

Otherwise, the adjoint state ODEs have to be integrated numerically."

I guess most real life models contain conserved quantities, making the Jacobian singular Jacobian is non-singular. I am not conviced that a speedup of 4.4 justifies all these efforts.

**Have the authors made all data and (if applicable) computational code underlying the findings in their manuscript fully available?**

Reviewer #1: Yes

Reviewer #2: Yes

Reviewer #3: Yes

PLOS authors have the option to publish the peer review history of their article (what does this mean?). If published, this will include your full peer review and any attached files.

Reviewer #1: No

Reviewer #2: No

Reviewer #3: No
---

## [Decision Letter · Decision Letter 1]

1 Dec 2022

Dear Lakrisenko,

We are pleased to inform you that your manuscript 'Efficient computation of adjoint sensitivities at steady-state in ODE models of biochemical reaction networks' has been provisionally accepted for publication in PLOS Computational Biology.

Best regards,

Attila Csikász-Nagy

Academic Editor

PLOS Computational Biology

Mark Alber

Section Editor

PLOS Computational Biology

Reviewer's Responses to Questions

**Comments to the Authors:**

Reviewer #1: The authors have adequately addressed my concerns (and in my opinion those of the other Reviewers). I recommend publication.

Reviewer #2: The authors have made a thorough revision of the paper and addressed all the comments raised in the initial review. The answers are detailed and convincing. The efficient computation of gradients is really important in model analysis and the significant technical improvement made by the authors justifies publication in my opinion.

Reviewer #3: The authors properly answered all questions and comments.

**Have the authors made all data and (if applicable) computational code underlying the findings in their manuscript fully available?**

Reviewer #1: Yes

Reviewer #2: Yes

Reviewer #3: Yes

PLOS authors have the option to publish the peer review history of their article (what does this mean?). If published, this will include your full peer review and any attached files.

Reviewer #1: No

Reviewer #2: No

Reviewer #3: No

---

## [Editor Report · Acceptance letter]

28 Dec 2022

PCOMPBIOL-D-22-01244R1 

Efficient computation of adjoint sensitivities at steady-state in ODE models of biochemical reaction networks

Dear Dr Lakrisenko,

I am pleased to inform you that your manuscript has been formally accepted for publication in PLOS Computational Biology. Your manuscript is now with our production department and you will be notified of the publication date in due course.

With kind regards,

Zsofi Zombor
